# When the Relationship Is at Stake: Parents’ Perception of the Relationship with a Child with Problematic Gaming and Their Perceived Need for Support

**DOI:** 10.3390/healthcare12080851

**Published:** 2024-04-17

**Authors:** Marie Werner, Sabina Kapetanovic, Maiken Nielsen, Sevtap Gurdal, Mitchell J. Andersson, Alexandru Panican, Emma Claesdotter-Knutsson

**Affiliations:** 1Department of Child and Adolescent Psychiatry, Skåne University Hospital, 22185 Lund, Sweden; emma.claesdotter-knutsson@med.lu.se; 2Department of Clinical Sciences Lund, Faculty of Medicine, Lund University, 22100 Lund, Sweden; mitchell.andersson@med.lu.se; 3Department of Psychology, Stockholm University, 11418 Stockholm, Sweden; sabina.kapetanovic@hv.se; 4Department of Behavioral Studies, University West, 46132 Trollhättan, Sweden; maikendue@hotmail.com (M.N.); sevtap.gurdal@hv.se (S.G.); 5Malmö Addiction Center, Skåne University Hospital, 20502 Malmö, Sweden; 6School of Behavioural, Social and Legal Sciences, Örebro University, 70182 Örebro, Sweden; alexandru.panican@soch.lu.se

**Keywords:** problematic gaming, addiction, gaming disorder, parent–child relationship, parenting needs

## Abstract

Intrapersonal parental factors play a significant role in the development of problematic gaming in children. However, few studies have explored parental perspectives on their relationship with a child engaged in problematic gaming, as well as the need for support parents perceive in relation to the child’s gaming. We conducted semi-structured interviews with 12 parents (83.3% women) of 11 children (81.8% boys, Mage = 15 ± 2) to examine how parents of children with problematic gaming behavior perceive the parent–child relationship and their need for additional support. We analyzed qualitative accounts using thematic analysis to identify themes and subthemes while drawing on the theoretical frameworks of Aaron Antonovsky’s theory of sense of coherence (SOC) and Jürgen Habermas’ theory of logic. Participants described difficulties regarding all three components of SOC (meaningfulness, comprehensibility, and manageability) in relation to their child’s gaming, with the most significant challenge being manageability. Parents primarily sought assistance from institutions and organizations, such as mental health services, to enhance manageability. The findings emphasize parents’ need for relational and practical support tailored to their unique context, as well as their wish to be more involved in the treatment of their children.

## 1. Introduction

In 2018, the World Health Organization (WHO) introduced the 11th revised version of the International Classification of Diseases (ICD-11), which included gaming disorder (GD) as a medical condition characterized by impaired control and a significant preoccupation with gaming, where individuals prioritize it over other activities [1,2]. Similarly, the Diagnostic and Statistical Manual of Mental Disorders Fifth Edition (DSM-5) from 2013 included internet gaming disorder (IGD) as a provisional diagnosis needing further research [3]. While the prevalence of GD and IGD in Sweden is, to date, unknown, a European epidemiological study from 2015 reported that 5.1% of adolescents met up to four criteria for IGD [4]. With respect to comorbidity, it is well recognized that children with neurodevelopmental impairments (NDIs) are a group whose needs must be carefully considered in relation to the phenomenon of problematic gaming. For example, children with NDI, and boys in particular, are at risk for gaming disorder [5]. Although research has revealed positive effects for children with NDI, for example, children with autism spectrum disorder (ASD) report having more friends and feeling less lonely than non-gamers with ASD [6], it is also a higher risk for developing problematic gaming for individuals with ADHD compared to individuals without ADHD [7]. Studies have shown that there is some comorbidity for those who are at risk for gaming disorder or problematic gaming. For example, children with NDI spend more time on online games compared to children without NDI [5,7,8] and the design of the games that give instant gratification is appealing for those children [9].

Children and their behaviors are often derived from and embedded within a family ecosystem [10]. Thus, how parents deal with parenting challenges and how children cope with challenges, such as problematic gaming, may be related to the composition and quality of the relationship between parent and child. Indeed, research has demonstrated that adolescents with IGD may distrust and have poorer communication with their parents than adolescents who do not meet the diagnostic criteria [11]. In contrast, strong parent–child relationships can serve as a protective factor against developing IGD [12]. In that sense, the actions undertaken by parents in their role as caregivers influence their children’s behavior. It behooves parents and legal guardians that their child receives care for mental health problems when necessary. However, parents themselves are intricate individuals within the family framework, grappling with their own challenges and capabilities to navigate challenging situations. 

Research on the perspective of parents regarding their children’s problematic gaming is scarce. One study indicated that mothers of children with or without IGD have similar attitudes toward their children, yet mothers of children with IGD were more likely to report elevated depressive symptoms [13]. Despite a dearth of research related to IGD, studies outlining the parent–child relationship in the presence of substance use disorders may provide further insight. For example, parents of adolescents with opioid use disorder have described serving as their child’s case manager while juggling self-criticism, periods of isolation, and emotional strain [14]. Similarly, parents of adult children with drug addiction report experiencing feelings of guilt, shame, and social stigma [15]. These studies provide some insight into the challenges and experiences faced by parents of children struggling with addiction problems.

How parents deal with challenges in their parenting and their children’s healthcare may be explained through the theoretical lenses of Antonovsky’s theory of sense of coherence (SOC) [16]. According to the theory of SOC, individuals have the ability to handle challenges and stressors when they have a strong sense of comprehensibility (i.e., the individual’s perception of their ability to understand and predict events in their life), manageability (i.e., the individual’s perception of their ability to manage events with the resources and skills available to them), and meaningfulness (i.e., the perception that life, or events, have purpose and value). Importantly, all three components of SOC must be present to achieve a sense of coherence, that is, to perceive the ability to understand and manage life, put value in it, and successfully deal with challenges. In addition, Habermas’ theory of the life world and system world [17] may also help to operationalize the support needed for parents of children with problematic gaming. According to Habermas, there are two spheres of human experience: “the life world” and “the system world”. The life world is the sphere of everyday life where long-term relationships, like family and friendships, thrive with unconditional support. In contrast, the system world (institutionalized structures, organizations, and social systems) operates on instrumental logic, focusing on efficiency and control. Relationships within the system world are conditional and reflected in communication style, which consists of formal communication channels and protocols for sharing information. It includes economic and administrative structures that sometimes overshadow the sphere of the life world. Habermas uses the concept of colonization to describe how the life world is sometimes dominated by the logic of the system world. In the role of a relative or patient, one stands with a foot in the life world, where everyday life occurs, while contact with systems, such as healthcare, represents the system world with diagnostic classifications and standardized assessments. 

The present study aimed to examine qualitative narratives provided by parents regarding their perceptions of the relationship with their child who engages in problematic gaming, along with an assessment of their perceived need for support. The following research questions guided the study questions: (1)How do parents experience their relationship with a child who exhibits problematic gaming?(2)Do parents perceive a need for parental support relating to the child’s gaming, and if so, what form of support and from whom?

## 2. Materials and Methods

This study was conducted parallel to an ongoing research project assessing the efficacy of an individual cognitive behavioral therapy (CBT)-based treatment for problematic gaming in adolescents [18]. Both projects were carried out by the regional healthcare system’s outpatient clinics for child and adolescent psychiatry (CAP) in southern Sweden. CAP Skåne is responsible for providing psychiatric treatment to the inhabitants of Skåne county (1.4 million).

### 2.1. Participants

Participants were recruited from the aforementioned RCT using a convenience sampling strategy. The children in the RCT were all diagnosed according to prevailing standards within CAP. Their diagnoses were then validated by a senior consultant in child and adolescent psychiatry with over 10 years of clinical experience. While participants in this study were not explicitly asked about their children’s difficulties beyond problematic gaming, interviews revealed that several participants’ children suffered from NDI, such as ADHD and ASD. A total of 17 parents of adolescents who met a minimum of four of the proposed DSM-5 criteria for IGD and had undergone treatment were contacted by phone and invited to participate in this study. Three parents did not reply and three declined to participate or dropped out after the initial contact. In total, 12 parents (women = 10, men = 2) of 11 adolescents (boys = 9, girls = 2) were interviewed. In cases where more than one parent was listed, the parent who initiated contact with CAP was contacted. All parents were part- or full-time residential parents to the adolescent and considered themselves parents. One parent per child was interviewed, except on one occasion when both parents came to the interview as a result of a misunderstanding. After completing nine interviews, we observed a degree of saturation in the responses received [19]. However, two additional interviews were conducted to ensure sample information power. Table 1 displays a summary of the participant characteristics and the mode of the interview.

### 2.2. Procedure

Data were collected using semi-structured in-depth interviews [20]. The interview guide was developed from the study’s research question and included questions about family history and current situation, as well as the parent–child relationship before and after the treatment. Further, the interviews were conducted by the first author using an interview guide with open-ended questions. The interview questions encompassed different types, including introductory, exploratory, specifying, and direct questions [21]. The parents were interviewed by the first author two weeks to three months after the child had completed the treatment for problematic gaming. Participants were offered the choice between conducting the interview in person, by phone, or via video calls. All participants but one, who chose to interview by phone, opted to conduct the interviews via a video call. The interviews lasted between 25 and 60 min. This study was reviewed and approved by the Swedish Ethical Review Authority (DNr 2022-01289-02). Informed consent was obtained from all participants prior to partaking in this study.

### 2.3. Data Analysis

Interviews were digitally recorded and transcribed verbatim. In brief, we followed the six steps recommended by Braun and Clarke for thematic analysis [20]: (1) familiarization with the material, (2) systematically assigning codes to interesting features in the material, (3) identifying initial themes and collecting all data relevant to each theme, (4) reading through all themes concerning the coded segments and collected data, (5) defining and naming themes, and (6) presenting the results in a clear and structured manner. When the data were coded and presented, reflections on the theoretical link were discussed. We also checked for overlaps and used the research questions as a basis for the presentation together with the theoretical framework that we found suitable for our results. Many of the statements and themes were recognized in Antonovsky’s theory of SOC and Habermas’s theory of the life world and system world. Therefore, the data and continuation of the analyses were performed through these lenses. Extracts from the interviews were translated from Swedish to English by a bilingual researcher to provide a clearer picture of the themes.

## 3. Results

The analyses of the empirical data with the theoretical approach of Antonovsky and Habermas revealed several themes linked to this study’s aim. Theme 1 related to how the parents experience their relationships with their child who exhibits problematic gaming behavior and theme 2 related to parents’ perception of the support they seek for their child´s gaming. From each theme, we derived a set of subthemes (see Table 2).

### 3.1. Parents’ Description of the Parent–Child Relationship

The first theme revealed how the parents perceive the relationship between them and their children engaged in problematic gaming. Parents expressed that they support and protect their child but find it difficult to understand the child´s gaming habits. These expressions were divided into three subthemes: “The protective and supportive parent”, “The understanding parent”, and “The controlling parent”.

### 3.2. The Protective and Supportive Parent

This subtheme revolved around the emotional aspect of being a parent to a child with problematic gaming. Parents described a strong emotional commitment to helping their child cultivate healthier gaming habits, a sentiment that can be understood in terms of meaningfulness. One parent stated

[…] He and I have a very open relationship […] because he knows that I’m his soldier; I always stand on the front line and defend him in all situations against everyone because he is not capable of doing it himself. (Parent of Patient 10.)

The sentiment underlying the above quote suggests that it is worth trying to support or fight for your child, much like a soldier. Even if the situation with a child that has problematic gaming is described as difficult, none of the parents expressed any willingness to give up or that their effort to help is worthless. On the contrary, they exuded endurance and engagement in supporting their child. Furthermore, parents’ willingness to help their child sometimes took on a self-sacrificial form. Several parents spontaneously described their personal health challenges, including stress and anxiety stemming from concerns about their child’s well-being.

I am putting my own life on hold here until he comes of age […] there are so many fights I have to take with society and the authorities for him to get the support he is entitled to. […] I don’t mind picking up so much, maybe that’s why I’m burned out, absolutely, but he is still more important than myself. (Parent of Patient 10.)

Notably, these accounts consistently conveyed a clear subtext: the challenges they encountered were perceived as worthwhile because they contributed to the child’s well-being.

### 3.3. The Understanding Parent

The second subtheme is about how parents described their struggle to comprehend their child’s strong desire to game to such an extent that it adversely affected other aspects of life. Parents developed various interpretations to make sense of the situation. Some adopted a disease description of the behavior, while others related it to the child’s mental health and overall functioning. One parent tried to understand the problematic gaming as an addiction and said the following:

Let us call it a form of addiction because that is what it is. It gives your brain a kick, which is what you are after as someone with a gaming addiction. […] (Parent of Patient 4.)

Yet, others explicitly described gaming as a normal part of adolescent development.

He is very good at hiding it [the gaming behavior], so we thought, rebellious teenagers, they are like that; moody […] it is a process of emancipation from the parents (Parent of Patient 5.)

This statement describes problematic gaming as something that might be a process and will disappear when the child gets older, but then she continued and remarked the following:

It also turned out that he wasn’t sleeping either […] his brain was so agitated that he only slept for two to three hours a night. […] Then he couldn’t hold it together any longer and it broke down […] then we had to take him to the emergency room. (Parent of Patient 5.)

There were also parents who made efforts to comprehend the digital realm but lacked interest, while their child remained too withdrawn to allow their parent to get involved in their digital experiences. The parent might have tried to engage and learn about this world, but the child did not want to let them into their own sphere. One parent described it as difficult to enter the child´s domain.

It’s very difficult, I’m totally uninterested in the internet […] I don’t have the same need for it as other people […] but one day I said, ‘Well, I’ll sit on a chair behind you so I can be with you when you play’, and then she was like, ‘No, you really can’t be there!!’ *laughs* (Parent of Patient 7.)

Notably, none of the parents expressed an inability to understand how the problem emerged and had created their own understanding of the situation they found themselves in together with their child.

### 3.4. The Controlling/Monitoring Parent

While the participants derived meaning and, to some extent, created comprehensibility concerning their child’s problematic gaming, they grappled with their perceived ability to manage or take control of the problem. Participants described exploring a broad spectrum of material (e.g., limiting internet or computer access) and immaterial (e.g., attempting to reason with the child or using their authority as parents) strategies to reduce gaming. Additionally, parents revealed internal conflicts concerning manageability when they recognized the potential benefits associated with gaming:

A boy who has been incredibly lonely […] and is starting to blossom and thrive a bit. […] It brings joy to a mother’s heart when you see that. […] When you have all these years behind you where he sat alone, day in and day out, year after year, and never had a friend. That has been my big dilemma, where should I draw the line? […] (Parent of Patient 3.)

One strategy parents used to enhance manageability was to persuade their child to undergo treatment for their problematic gaming, either through negotiations with the child or by making it obligatory.

I have not said, ‘If you want to participate’, I have transformed it into ‘You will participate’ because I have noticed that when she is given tasks to do, she does them. But if an adult says, ‘If you want’, it is the same as saying, ‘Oh, then it is optional […]’ (Parent of Patient 7.)

Persuading the child to undergo treatment represents a concrete form of material resource utilization to enhance manageability. However, for many parents, the step to contact healthcare professionals was only taken when they perceived their own efforts and skills to be insufficient.

### 3.5. Parents’ Perceived Need for Support

Employing Habermas’ [17] theory of the life world and system world as a framework for assessing parents’ perceptions of the need for parental support concerning their child’s gaming, we identified what forms of support parents sought and the primary sources of this support. Participants’ reflections could be neatly categorized according to theory, resulting in the formation of two subthemes. The first one pertains to how parents sought care from outside family, referred to as “Seeking support from the System World”, which includes interaction with, for example, healthcare professionals. On the other hand, the second subtheme, labeled “Seeking Support from the Life World”, encompassed interactions with family, friends, and others in similar situations. Some parents described a struggle to balance the support they received from both worlds.

### 3.6. Seeking Support from the System World

In the first theme, we noticed that parents articulated that manageability regarding their adolescents’ problematic gaming was their greatest challenge. Many initially made efforts on their own, implementing various rules and strategies, before seeking help from representatives of the system world (i.e., healthcare professionals). Parents held the belief that the system’s knowledge and representatives would be better equipped to improve the situation.

[…] you are generally not professionally trained, at least not as a parent, to handle someone who has, let’s call it a form of drug addiction. […] and since we have dealt with him for perhaps too long and maybe should have taken action and requested more help earlier. (Parent of Patient 4.)

Parents mainly expressed a desire for concrete guidance from healthcare professionals. Specifically, they sought information regarding how many hours of gaming are optimal for their child, guidance on balancing accommodations for the child’s difficulties versus setting demands, and insights into the appropriateness of restricting access to the internet. One parent had, for example, attended parent support classes to better understand how to parent their child. She described the following:

I have been driving this myself because I have felt that I need to have as many tools as I can, just to cope, to have a functional life. So I have definitely participated in parenting courses organized by the habilitation service. It hasn’t been specifically about gaming, but it’s been more general, and about developing my toolbox, how to deal with situations and so on. (Parent of Patient 3.)

This quote illustrates parents’ need for support and their efforts to learn how to be a better parent and handle their child’s problematic gaming behavior. However, other parents doubted or were unsatisfied with the help and support they received from peers and healthcare professionals. One parent said the following:

People have so many opinions. When I talk to some people, I get a lot of support for the way I think, but at the same time, it’s because they have no idea how much he needs. So I don’t have anyone to talk to about this either. […] I would have needed more support in what is reasonable […], perhaps also from the habilitation service. […] Now we have still heard a lot of “it’s just a lot of teenage years”, and I recognize that, […] but even if he is a teenager, you should still brush your teeth. […] I don’t think so, I don’t agree, that’s what I can say, I don’t feel it’s right, so I, well... I feel a little alone in this. (Parent of Patient 9.)

This parent implies that she knows her child best and believes that certain advice may not be applicable to her child, resulting in conflicting thoughts about what is appropriate or not. A common thread among parents was the sense that they had to take on the role of their child’s case manager. They described engaging with various societal actors, including healthcare professionals, social services, and rehabilitation staff, to enhance support for their child. Parents often found themselves acting as intermediaries, relaying information between various societal agencies.

We have established contacts where we are the spider in the web and have to push for the smallest things all the time. […] It could be so much smoother if we did not have to inform CAP about what the rehab center is doing, social services about what CAP is doing, and the school about what the rehab center is doing, and so on, without the authorities talking to each other. (Parent of Patient 4.)

Furthermore, parents conveyed a belief that a greater level of parental involvement in the child’s treatment could have resulted in more favorable treatment outcomes for the child and the family as a whole.

I did not receive any information from her [the practitioner] about what had happened […] I did not receive any information about what they had talked about. If I had known what topics she brought up, I would have certainly talked to him about it in the evening. If a child develops a gaming addiction, it is not just the child’s problem; it affects the whole family, and then you cannot exclude the parent living in the conflict. […] (Parent of Patient 11.)

Despite some participants having reservations about their level of involvement, they did not appear to have questioned or opposed this setup. Instead, they adapted. This adaptation can be seen as a response to the system’s rules, even if it comes at the cost of their own preferences and everyday experiences. Parents requested concrete advice, yet they also desired active involvement and to have their voices heard by healthcare professionals concerning their child’s treatment. They emphasized that they are the only ones who have the whole picture and, thus, need to be part of the care the child receives.

### 3.7. Seeking Support from the Life World

The system world is governed by facts, regulations, and guidelines, which were values sought after by parents. Yet, parents also expressed a need for relational support, albeit to a lesser extent. This is the second subtheme and shows that often, the closest agent from the life world is the child’s other parent, whom the parent must cooperate with to raise the child. However, several parents had different views on how to best support their child, and they believed that the system’s representatives could help them address these issues. Conversely, some participants described receiving enough support from the other parent or partner to fulfill their need for relational support.

We have talked a lot, her father and I, because she alternates living between us, to have the same rules and approach. We have tried to have that. And then my boyfriend has older children, one of whom has played a lot, so I have had support and talked to him about how to handle it. (Parent of Patient 11.)

There are also parents who found it difficult to handle their child´s gaming problems since they do not have the same ideas about how to deal with them. For example, one parent said the following:

My husband and I don’t agree on which battles are worth fighting. I believe I have a better understanding of his energy and his issues. […] My husband thinks I’m his slave. He doesn’t think I set boundaries. […] He has no respect for his father because I always step in and fix things—we’re not on the same page there, but we’ve started family therapy [with social services]. (Parent of Patient 10.)

This discrepancy sometimes made parents seek support from others. For example, one parent expressed desire to be in contact with others who had similar experiences.

[…] I feel incredibly lonely, and since it’s quite a stigmatized subject, you know. Everyone always wants to talk about how well their children are doing and how they succeed and such […] If one had met with other parents in the same situation, maybe one would have understood it was, like, [a] problem, that existed. (Parent of Patient 7.)

Engaging with other parents navigating similar challenges can serve to fulfill relational support needs without the burden of judgment or shame.

## 4. Discussion

Our study examined parents’ perceptions of their relationship with a child engaged in problematic gaming and their perceived need for support. Through this study, we have expanded our understanding of parents’ experiences concerning their relationship with their own child struggling with problematic gaming, as well as the support they have sought and desired, something that has not previously been investigated in the field.

### 4.1. Parents’ Perception of the Parent–Child Relationship

Antonovsky [16] described meaningfulness as the driving component in SOC theory. Indeed, our findings indicated that parents’ emotional commitment and sense of purpose mobilized them to manage the strains associated with parenting a child with problematic gaming, even though this commitment can negatively impact their mental well-being, as seen in previous research [13]. Concerning comprehensibility, parents in our study adopted various explanations to make sense of their child’s gaming behavior. Some developed a disease description, whereas others linked it to mental health, overall functioning, or typical adolescent development. While comprehensibility, according to Antonovsky [16], can empower effective coping and enhance problem-solving abilities, it can also delay problem recognition and intervention. For instance, if parents view their child’s gaming behavior as a natural part of adolescent development, it might take them longer to recognize it as a problem that requires attention and intervention. Future studies should explore how comprehensibility affects the development and treatment of adolescent problematic gaming. Manageability involves parents’ perception of their ability to handle their child’s gaming behavior with available resources. Despite parents’ attempts to influence or control their child’s gaming, they reported feeling a lack of control. According to SOC theory, all three components must be present to fully cultivate a sense of coherence. Our findings suggested that parents in this study did not fully experience a sense of coherence. Despite being emotionally committed, they lacked the resources necessary to guide their child’s gaming habits and reported mental health struggles themselves. These findings may be comparable to studies on parents of children with substance use disorders, who reported emotional strain when dealing with their child’s opioid addiction [14].

### 4.2. Parent’s Perceived Need for Support

Addressing the second research question, parents recognized their need for support from both the system world (i.e., healthcare professionals) and the life world (i.e., community and personal relationships). Primarily, their support needs aligned with the logic of the system world [17], as they sought explicit instructions and guidance from healthcare practitioners. However, the system’s capacity to provide personalized, success-oriented guidance was limited. While healthcare experts possess a wealth of knowledge, their operation within structured routines, protocols, and division of responsibility can pose challenges in meeting the parent’s exact support needs. Notably, the aspiration to expand the system’s capacity and receive more individualized and holistic treatment resonates with other studies investigating caregivers’ support needs [14]. The parents’ attempt to reconcile the logic of the life world with the logic of the system world resulted in frustration, as they felt that they were forced to adapt to the logic of the system world. Thus, this adaptation can be seen as a colonization of the system’s logic at the expense of the life world.

Furthermore, parents sought support from the life world, albeit to a lesser degree. Among those who reported having a supportive co-parent or partner, the need for additional support from the life world was less pronounced. Conversely, parents who lacked sufficient relational support experienced emotions of frustration and isolation. Additionally, there was a shared desire among parents to connect with others who were navigating similar challenges. As parents of children with substance or behavioral addictions often experience self-criticism and periods of isolation and emotional strain [14], the inclination to seek out other parents in similar situations can be interpreted as a quest for a safe space to express feelings without fear of judgment [22].

In seeking a balance between the life and the system worlds, participants described their frustrations with the lack of communication and coordination among agencies involved in their child’s care. They advocated for system representatives to possess a more comprehensive grasp of the child and their treatment. Despite being unsatisfied with the rules and structures that govern public healthcare practices, the participants adapted rather than questioned them. Instead, they assumed the role of intermediaries between various agencies, even though they doubted this arrangement’s effectiveness in addressing their family’s needs. These experiences highlight the challenges parents face in reconciling the regulations and expectations of the system world with the intricacies of their everyday lives.

As shown above, the conclusions of this study are in line with the results of previous studies carried out in the field of addiction. The findings strengthen the assumption that the problem of problematic gaming in children can be understood from an addiction perspective and that parents of these children can experience similar challenges as parents of children with, for example, substance abuse.

## 5. Limitations

This study has limitations that should be considered, and the results should not be interpreted as universally generalizable. Firstly, the participants’ children presented a range of comorbid mental health issues alongside problematic gaming. The participants were not explicitly asked about their child’s potential diagnoses. However, given that participants were recruited through the previously mentioned RCT [18] in which the most common coexisting conditions were ADHD, ADD, ASD, and depression, it is reasonable to assume that these conditions may have been present among participants’ children.

Given the intertwining nature of problematic gaming with other mental health aspects, it is important to recognize that this study may reflect broader mental health issues rather than solely focusing on problematic gaming. Despite this challenge, the empirical data and study design still offer valuable insights, which are particularly relevant in clinical contexts for treating IGD or problematic gaming, considering their frequent comorbidity in child psychiatry [23].

Another limitation pertains to the potential influence of the interview guide used. Compared to an unstructured interview, the guide may have directed the conversations toward the predetermined themes, possibly overlooking valuable knowledge that participants might possess [24]. Nevertheless, employing an interview guide does offer certain advantages, such as facilitating structure and allowing for greater control over the topics addressed during the interview, which enables comparison across participants’ responses. Moreover, this study has a small sample and utilizes a qualitative approach. Thus, while our design provides in-depth insight from the parents’ perspective, our findings may not be generalizable to the experiences of parents from the general population since more than 83% of the study participants were female respondents, and previous research has demonstrated that gender is a significant variable when examining the relationship between children’s gaming and their relationships with their parents [25]. Despite this, the findings can be used to advance the development of treatments for problematic gaming with a family perspective by incorporating parents.

## 6. Conclusions

The present study explored parents’ perception of the parent–child relationship with a child engaged in problematic gaming and the parents’ perceived support needs. Firstly, parents expressed a strong sense of purpose in supporting their child in engaging in balanced gaming habits, while crafting personalized explanations to make sense of their circumstances. However, many parents reported that they lacked the resources or skills to effectively support their child in developing better gaming habits. Parents primarily sought support through instrumental logic, such as practical advice and specific guidelines, yet expressed frustration with the healthcare systems’ inability to provide individualized guidelines tailored to their specific needs. Furthermore, parents sought support from partners, co-parents, and parents of children with similar problems. These findings offer valuable insights into parents’ experiences and support needs but also raise further questions about addressing problematic gaming in a manner that supports both the child and their family. These findings can be used to develop and implement treatment methods that enhance parents’ sense of manageability by providing them with better resources and finding ways to involve them more in their child’s treatment.

## Figures and Tables

**Table 1 healthcare-12-00851-t001:** Descriptives of participants in this study.

Patient-ID	Parent Gender	Child Gender	Child Age	Interview Type
1	Woman	Boy	14	Video call
2	Woman	Boy	17	Phone call
3	Woman	Boy	16	Video call
4	Man	Boy	16	Video call
5	Woman	Boy	17	Video call
6	Woman	Boy	18	Video call
7	Woman	Girl	14	Video call
8	Woman and man	Boy	14	Video call
9	Woman	Boy	15	Video call
10	Woman	Boy	12	Video call
11	Woman	Girl	12	Video call

**Table 2 healthcare-12-00851-t002:** Themes and subthemes.

Themes	Subthemes
1. Parents’ description of the Parent–Child Relationship	The protective and supportive parent The understanding parent The controlling parent
2. Parents’ Perceived Need for Support	Seeking support from the system world Seeking support from the life world

## Data Availability

The data presented in this study are available on request from the corresponding author.

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
