# Peer review of "When the Relationship Is at Stake: Parents’ Perception of the Relationship with a Child with Problematic Gaming and Their Perceived Need for Support"

_healthcare, 2024, doi:10.3390/healthcare12080851_

Round 1

Reviewer 1 Report

Comments and Suggestions for Authors

This is a wonderful topic but the paper needs to improve across the board - language, more research into introduction and conclusions, did you explore the idea of neurodiversity, such has undiagnosed ASD in these children and their social difficulties? 

Comments on the Quality of English Language

must be improved, too many grammatical errors and language doesn't flow naturally 

Author Response

Thank you for your time to read our manuscript. We have made adequate changes to the manuscript and believe this has improved it. We have added extra research to the introduction section and have carefully read through the script as a whole. We think our conclusions are aligned with the results of this study and specifically what parents express in term of their parenting and their parenting needs

One of the authors are native American and has gone over the manuscript again to correct the grammatical and spelling errors that unfortunately occurred. We hope that you experience the language as improved.

The children in the study were all diagnosed according to prevailing standards within CAP. Their diagnoses were validated by a senior consultant in child and adolescent psychiatry with over 10 years of clinical experience. 

Marie Werner and colleagues

Reviewer 2 Report

Comments and Suggestions for Authors

Dear Authors,

Thank you for letting me review this interesting and important paper: When the relationship is at stake: Parents' perception of the relationship with a child with problematic gaming and their perceived need for support.

 My suggestions:

·        Discuss in the limitations that 84% of the parents were women.

·        Delete and on page 2 Line 46.

·        Delete space page 3 line 103.

·        According to themes and subthemes are the verbs' description and perceived, correct? Page 4 line 160, you write parents perceived.

·        Why do you not include quotes from participants 1, 2 and 6?

Author Response

Thank you for your valuable comments. We have made adequate changes in the manuscript believe that this has improved it. We have added a phrase in the limitations section regarding the gender of the parents.

Thank you for pointing out the incorrectness of the verbs “description and perceived” we have changed this accordingly.

As you correctly noted are not all participants quoted in the text. The quotes that were selected were chosen because they concisely represent the central themes of the study and most clearly summarize similar responses from the participants.

Marie Werner and colleagues

Reviewer 3 Report

Comments and Suggestions for Authors

This study is relevant for publication. Firstly, it fills a gap in the literature by examining parents' perceptions of their relationship with children who have gaming problems, thereby contributing to the understanding of problematic gaming in children and adolescents. Additionally, the results have important implications for clinical practice, providing insights that can inform more effective interventions and treatments for this issue, especially in child mental health contexts.

The detailed qualitative approach employed in this study allows for a deeper understanding of participants' experiences, capturing nuances that could be missed in quantitative methods. This results in richer and more contextualized insights into the topic.

Author Response

Thank you for reading our manuscript. We have made changes according to the comments from the reviewers and believe that the manuscript has improved, we hope that you agree.

Marie Werner and colleagues

Reviewer 4 Report

Comments and Suggestions for Authors

I have read the article with great interest, since it offers a fresh viewpoint on the issue and is based on a compelling theoretical framework. The text addresses a significant and imperative social topic. The research goal was defined in a relatively clear manner, so was the selected research method, which merits particular given the qualitative method is less frequently used in studies on addiction, while it is valuable. As a result, I think the selection of the research methodology was excellent. Unfortunately, I feel like there is a gap in this regard: the article's content failed to inform the reader of the procedures used to choose test subjects for the study, i.e. how participants were recruited. I would expect this type of information essential in the text. I believe in-depth interview technique was employed, as in-depth interviews are generally conducted on small samples, with purposive sampling, and thus, they cannot be generalized to total population. It should be noted explicitly as well that papers derived from in-depth interviews frequently include respondents' exact quotes, which offer unique insight into their perspectives. It is important from the point of view of a reader though that the text offers a sound summary of the whole gathered material, and is not merely a selected of quotes. I would recommend clearly emphasizing this in the text.

I appreciate that the authors are conscious of the limits and, appropriately, refrain from extrapolating the study's findings to the wider population in favor of concentrating on the analysis of the group under investigation.

Regardless of my comments, I must admit that the article is structured correctly, with a compelling story and concise language that is easy to understand.

Finally, I would like to emphasize that the learnings from the studies presented in the article should serve as a starting point for further research.

Author Response

Thank you for reading our manuscript. We have re written the Method and Introduction section and hope that this has improved the overall quality of the manuscript.

Marie Werner and colleagues